# Inhibition of Inflammatory Regulators for Chronic Obstructive Pulmonary Disease (COPD) Treatment from Indonesian Medicinal Plants: A Systematic Review

**DOI:** 10.3390/cimb47040262

**Published:** 2025-04-08

**Authors:** Dyah Iswantini, Min Rahminiwati, Trivadila Trivadila, Siti Sadiah, Novriyandi Hanif, Rut Novalia Rahmawati Sianipar, Susi Indariani

**Affiliations:** 1Department of Chemistry, Faculty of Mathematics and Natural Sciences, IPB University, Bogor 16680, Indonesia; trivadila@apps.ipb.ac.id (T.T.); nhanif@apps.ipb.ac.id (N.H.); rutnovaliasianipar@apps.ipb.ac.id (R.N.R.S.); 2Tropical Biopharmaca Research Center, IPB University, Bogor 16128, Indonesia; minrahminiwati@gmail.com (M.R.); sitisa@apps.ipb.ac.id (S.S.); susiin@apps.ipb.ac.id (S.I.); 3School of Veterinary Medicine and Biomedical Sciences, IPB University, Bogor 16680, Indonesia

**Keywords:** COPD, Indonesia, inflammation, natural resources, systematic review

## Abstract

A characteristic of Chronic Obstructive Pulmonary Disease (COPD) is the inflammation of the respiratory pathway. Inflammatory regulators affected in COPD include cyclooxygenase-2 (COX-2), tumor necrosis factor-alpha (TNF-α), interleukin (IL-1β), and interleukin-6 (IL-6). Therefore, this systematic review aimed to investigate the biological activity of Indonesian medicinal plants as anti-inflammation through in vitro, in silico, and in vivo studies. A digital search was conducted using Scopus, Google Scholar, and PubMed online databases to find relevant articles by applying specific keywords related to the subject of interest. The results showed 18 studies that reported five Indonesian medicinal plants proven to inhibit inflammation regulation. The five plants were *sambiloto* (*Andrographis paniculata*), *legetan warak* (*Adenostemma lavenia*), *kersen* (*Muntingia calabura*), *babadotan* (*Ageratum conyzoides*), and *sembung rambat* (*Mikania micrantha* Kunth). In vitro studies, *A. paniculata*, *A. lavenia*, and *M. calabura* can suppress the levels of pro-inflammatory cytokines, such as TNF-α, IL-1β, and IL-6, in LPS-stimulated RAW 264.7 cells. In silico studies, compounds that have strong binding to bind inflammatory receptors are andrographiside contained in *A. paniculata*; ent-11α-hydroxy-15-oxo-kaur-16-en-19-oic acid contained in *A. lavenia*; 7-Hydroxyflavone contained in *M. calabura*; and 22,23-Dihydrospinasterol contained in *A. conyzoides*. In vivo studies, extracts of *A. paniculata*, *M. calabura*, *A. conyzoides*, and *M. calabura* can reduce inflammation in lung tissue in animal models (hamsters, mice, and rats). This systematic review might help to develop COPD treatment and build scientifically natural products from Indonesian medicinal plants for future investigations.

## 1. Introduction

Coronavirus disease 2019 (COVID-19) is an infectious disease caused by severe acute respiratory syndrome coronavirus-2 (SARS-CoV-2). This started in China and led to the 2019–2020 coronavirus pandemic in nearly all regions of the world, including Southeast Asia, Africa, America, the Eastern Mediterranean, Europe, and the Western Pacific. The sufferers can experience fever, dry cough, and serious symptoms such as respiratory failure and even death. Moreover, WHO reported, in February 2025, that an accumulated 777 million people were affected and more than 7 million deaths were recorded [1]. COVID-19 is associated with the respiratory system due to initiating acute respiratory distress syndrome (ARDS) and Chronic Obstructive Pulmonary Disease (COPD) when left untreated. COPD is a chronic inflammatory lung disease that causes airflow obstruction [2,3,4,5,6,7,8].

The effect of exposure to harmful particles and gases, especially those from cigarette smoke, will cause an inflammatory immune response or chronic respiratory inflammation in the lungs. This inflammatory response is responsible for the pathophysiology of COPD [9,10,11]. In addition, risk factors for COPD are indoor pollution, outdoor pollution, genetic factors, history of recurrent respiratory infections, gender, age, alcohol consumption, and lack of physical activity [12,13,14].

Several efforts are carried out to manage and treat patients with COPD, such as using oral, anti-inflammatory drugs(corticosteroids, and bronchodilator). However, long-term administration of corticosteroid drugs can cause side effects in the form of diabetes, osteoporosis, cataracts, and increased disease risk [13,15,16,17].

The number of side effects from corticosteroid drugs has triggered people to use herbal medicines. Pharmacological effects and relatively less toxic effects are often experienced from herbal medicines compared to synthetic drugs [18,19]. The trend of back to nature lifestyle and the use of natural/herbal medicinal materials is increasing to 88%. Additionally, the WHO has recommended the use of herbs in public health maintenance. Currently, the herbal market in the world and Indonesia is very popular as well as promising. The global herbal market in 2021 was USD 165.66 billion and is estimated to reach USD 5 trillion by 2050 [20,21].

The use of herbal plants in Indonesia is quite widely applied considering that Indonesia is a mega-biodiversity country. The stems, roots, petals, or leaves of medicinal plants are frequently decocted by Indonesians to prepare *jamu* or traditional herbal drinks. However, research on the treatment of COPD using Indonesian herbs is still limited. Our research team is currently focusing on research on this object; so, most of the results of this systematic review come from our research. Therefore, the inclusion criteria in this systematic review are as follows: Indonesian medicinal plants that can be used as COPD treatment that have been proven in in vitro, in silico, and in vivo studies. Data from reviews, case reports, and meta-analyses are not reviewed. We also linked this COPD treatment to COVID-19 because these diseases have similarities in inflammation in the respiratory pathway. We found five Indonesian medicinal plants for COPD treatment: *sambiloto* (*Andrographis paniculata*), *legetan warak* (*Adenostemma lavenia*), *kersen* (*Muntingia calabura*), *babadotan* (*Ageratum conyzoides*), and *sembung rambat* (*Mikania micrantha* Kunth). The five plants have also been compared in terms of the characterization of their compounds in other medicinal plants that have anti-inflammatory properties or the binding strength of the compounds to inflammatory receptors, which have been carried out by other researchers. The data we have obtained through this systematic review are expected to provide new insights into developing scientific herbal medicine in Indonesia and optimizing Indonesia’s natural resources.

## 2. Materials and Methods

The Preferred Reporting Item for Systematic Reviews and Meta-Analysis (PRISMA) guidelines were used to conduct the present review [22]. This systematic review was registered under PROSPERO (International Prospective Register of Systematic Reviews) with ID CRD420250645213.

### 2.1. Search Strategy

We performed a computer-based search in the principal databases, such as Scopus, Crossref, Google Scholar, PubMed, and Science Direct, related to inflammation in COPD and the treatment by Indonesian medicinal plants through in vitro, in vivo, and in silico studies. We collected publication period from 2001 to 2024 using the Publish or Perish software program version 8.17.4863.9118 (Windows GUI edition), and then we visualized the keyword relationship in the papers using VOSviewer software version 1.6.20 (Leiden University, The Netherlands). The visualization of structures was generated from the PyMOL software program version 3.0.3 (Schrödinger).

### 2.2. Inclusion and Exclusion Criteria

In the present systematic review, databases were selected to explore the published studies using the following keywords: “Chronic Obstructive Pulmonary Disease”, Indonesian medicinal plants”, “inflammation regulation”, “experimental studies”, “animal studies”, and “molecular docking studies”. The articles we obtained were then checked. We eliminated duplicate articles and reviewed whether the studies met the inclusion and exclusion criteria. The inclusion and exclusion criteria can be observed in Table 1. After that, we retrieved the full texts of the possibly pertinent studies to verify their eligibility. The comparators include Indonesian medicinal plants used in the treatment of inflammation and COPD; medicinal plants that have also been studied by other researchers; and data analysis that has similarities with the inflammatory activity of other medicinal plants. We reached 18 eligible studies to review. The flowchart depicted in Figure 1 shows the selection process of the studies.

## 3. Results and Discussion

### 3.1. Bibliometric Analysis

Research related to the inhibition of cytokine proinflammation and mediators of inflammation derived from medicinal plants shows the potential of compounds that are useful for reducing the activity of tumor necrosis factor-alpha (TNF-α), interleukin-1β (IL-1β), interleukin-6 (IL-6), and cyclooxygenase-2 (COX-2) and we can observe, in Figure 2A, notably, in Table 2, the relationship between inflammation, COPD, and COVID-19 in the same clusters, specifically, clusters 1 and 3. Figure 2B also clearly illustrates that comprehensive scholarly research on Chronic Obstructive Pulmonary Disease (COPD), inflammatory activities, and Indonesian medicinal plants was conducted extensively between 2020 and 2022, especially concerning natural products. In addition, there has not been much research on Indonesian medicinal plants in the treatment of COPD (Figure 2C). This happens because the color of “Indonesian medicinal plants” is green, which means there is a small amount or density of research on this item, but if the item color is yellow, it will indicate that the item has a higher research density. Therefore, this systematic review aimed to investigate several Indonesian medicinal plants that can inhibit the action of pro-inflammatory cytokines as COPD treatment agents and have been proven in vitro, in silico, and in vivo.

### 3.2. The Connection Between COPD and Inflammation

Based on statistical projections, COPD will be the cause of 4.4 million annual fatalities by 2040, with approximately 90% of deaths occurring in low- or moderate-income countries [20]. COPD conditions are divided into chronic bronchitis and emphysema. Figure 3 presents the difference between a normal lung and a lung affected by COPD. The lung affected by COPD has two phenotypes, including emphysema and chronic bronchitis. Chronic bronchitis has a relationship with airway inflammation, and the primary characteristic is chronic persistent coughing. This causes goblet cells to overproduce and mucus hypersecretion, restricting airflow. Meanwhile, emphysema initiates alveolar damage and the symptoms include dyspnea. Due to hyperinflammation and oxidative stress, emphysema pathogenesis consists of alveolar septa degradation, an increase in air space, and the loss of elastic recoil [23,24,25]. This respiratory condition frequently originates from biomass smoke exposure and smoking, which continuously obstructs airflow as well as impairs lung function and breathing. Inflammation is an important factor in the pathophysiology of COPD and several types of cells participate in the inflammation. However, the three most significant immune cell types are neutrophils, lymphocytes, and macrophages, which release cytokine pro-inflammation agents including TNF-α, IL-1β, IL-6, and IL-8. The release affects respiratory tract inflammation and causes lung tissue damage in airway disease and emphysematous destruction [26,27,28,29,30,31,32].

IL-1β and TNF-α are pro-inflammatory cytokines secreted mostly by macrophages. These trigger the production of matrix metalloprotein-9 and extracellular matrix protein-tenascin by macrophages as well as bronchial epithelial cells, contributing to emphysema pathophysiology. TNF-α promotes and activates the transcription factor NF-κB, known as the “kap-pa-light-chain-enhancer” of activated B cells, which regulates immunity, cell proliferation, and apoptosis. Additionally, IL-6 binds to particular membrane-bound IL-6 receptors existing only on hepatocytes and leukocytes. The receptors initiate an intracellular signaling cascade through the membrane-bound glycoprotein 130 (gp130) present on the membranes of numerous cell types and capable of triggering trans-signaling cascades. Glycoprotein 130 activation phosphorylates Janus-activated kinase (JAK), which stimulates several signaling pathways crucial to the immune response. The intracellular transcription of specific target genes related to the immune response is caused by the activated MAP (mitogen-activated protein) kinase pathway and the similarly activated JAK-STAT (signal transducers and activators of transcription) signaling pathway [29,33,34,35].

The COX-2 inflammatory pathway can be inhibited in COPD treatment. The rate-limiting enzymes in the metabolic process include cyclooxygenase (COX), which converts arachidonic acid (ARA) into prostaglandins (Figure 4). COX-2 is an inducible isoform of COX activated by various cytokines, stimuli, and tumor promoters. A study shows that acrolein, a component of cigarette smoke, promotes COX-2 production in rat lung epithelial cells through the NF-B pathway. Meanwhile, ARA is a highly abundant lipid metabolized by three families of enzymes—cyclooxygenase (COX), lipoxygenase (LOX), and cytochrome P450 (CYP) epoxygenase enzymes (CYP2C and CYP2J subfamilies)—into biologically active eicosanoid. The expression of COX-2 significantly increases during COPD development, and inhibiting the expression alleviates airway inflammation as well as remolding [36,37,38,39,40,41,42].

### 3.3. Characteristics of Included Studies

Table 3 shows the general characteristics of the 18 selected studies. Including the author, aims of the study, methodology, and overall findings.

### 3.4. In Vitro Studies

The cell viability test is the first procedure often performed in in vitro studies to ascertain the concentration of non-cytotoxic substances. This method is based on measuring cell viability in cell culture to determine the effects of pharmaceuticals in vitro on cell proliferation by cell-mediated cytotoxicity assays. Moreover, the cell viability test is generally classified into five types, including colorimetric, luminometric, fluorometric, dye exclusion, and flow cytometric assays. The most frequently conducted cell viability test is using the MTT assay originally defined by Mosmann [61]. The principle of this method is to measure the mitochondrial dehydrogenase activity of living cells which can convert MTT (3-(4,5-dimethylthiazol-2-yl)-2,5-diphenyltetrazolium bromide) into formazan. MTT reagent reacts with living cells and forms purple formazan crystals. In this context, tetrazolium succinate reductase (succinate dehydrogenase), an enzyme that integrates the respiratory chain in the mitochondria of live cells, reduces yellow tetrazolium salt to formazan crystals (Figure 5). Furthermore, the insoluble formazan salt is dissolved by adding solubilizing agents, and the colored result is quantitatively identified at 570 nm using a spectroscopic multiplate reader. The intensity of the purple color of formazan is measured as an absorbance value which is proportional to the number of living cells. The higher the intensity of the purple color formed, the higher the absorbance value obtained. This shows that more formazan is formed when many cells are alive and can react with the tetrazolium salt [62,63,64,65].

Lukito et al. [43] observed that a higher concentration of pure ethanol extract of *sambiloto* (*A. paniculata*) led to a lower percentage of cell viability in RAW 264.7 cells. The highest test concentration with 100% viability was found using the linear equation of pure ethanol extract (y = −802x + 131.84), generating an antilog x value of 40 μg/mL. This dose was assessed for the ability to suppress pro-inflammatory cytokines. To investigate pro-inflammatory cytokine inhibitory activity, TNF-α, IL-1β, and IL-6 levels were measured in LPS-stimulated RAW 264.7 cells using ELISA. LPS stimulation in positive controls can elevate TNF-α, IL-1β, and IL-6 by approximately 84.25% compared to negative controls. The pure ethanol extract of *A. paniculata* inhibited 100.00 ± 0.00% of TNF-α and IL-1β at a 40 µg/mL concentration, then reduced the activity of IL-6 by 85.59 ± 3.41%.

Astuti [44] investigated the percent viability of RAW 264.7 cells from a 70% water and ethanol extract of *legetan warak* (*A. lavenia*). The water extract of *A. lavenia* at concentrations of 12.5 μg/mL and 100 μg/mL triggered cytotoxicity because the resulting cell viability value was less than 60%, while, at concentrations of 25 μg/mL and 50 μg/mL, the cell viability values were 66.572% and 65.434%, respectively. Based on the cell viability values obtained, the *A. lavenia* water extract with a concentration of 25 μg/mL had the lowest cytotoxicity compared to other concentrations. The 70% ethanol extract of *A. lavenia* at a concentration of 50 μg/mL had the lowest cytotoxicity compared to other concentrations because the resulting cell viability was 73.186%. The cell viability values of 70% water and ethanol extracts with concentrations of 25 and 50 μg/mL showed that the extract was safe for RAW 264.7 cells. In the IL-6 inhibition assay, RAW 264.7 cells were only induced by lipopolysaccharide (LPS). The function of LPS is to stimulate inflammation for RAW 264.7 cells to release IL-6. The *A. lavenia* water extract at concentrations of 25 µg/mL and 50 µg/mL produced IL-6 levels of 14.594 µg/mL and 5.023 µg/mL, respectively, but the IL-6 inhibition value was lower than the control. After measuring the levels of IL-6 in LPS-induced cells and using *A. lavenia* extract, the effects of the extract on the viability of inflammatory cells were examined. The addition of water extract with concentrations of 25 μg/mL and 50 μg/mL to LPS-induced cells provided viability values of 83% and 75%, respectively. Meanwhile, using 70% ethanol extract with different concentrations (25 and 50 μg/mL) produced the same cell viability value of 67%. The obtained results showed that water extract with a concentration of 25 μg/mL was safe for the growth of cells experiencing inflammation when compared to control cells (83%).

Other studies on *A. lavenia* were carried out by Maeda et al. [45] and Kobayashi et al. [46], which were included in this systematic review. These showed that the compound *ent*-11α-hydroxy-15-oxo-kaur-16-en-19-oic acid played a role in inhibiting NO or served as an anti-inflammatory.

Furthermore, Tuwalaid et al. [47] evaluated the ability of *legetan warak* and *kersen* (*Muntingia calabura*) to inhibit COX-2. The research was performed in vitro with the ELISA method using the COX Inhibitor Screening Assay Kit and following the Cayman Chemical Catalog No. 701080 protocol. At a dose of 1000 μg/mL, the 96% ethanol extract of *legetan warak* inhibited 98.47% of the COX-2 activity, while 70% and 96% ethanol extract of *kersen* leaves inhibited 100% of the COX-2 activity. In another study, Lin et al. [48] discovered that an ethanol extract of *M. calabura* fruit (100 µg/mL) reduced COX-2 levels in RAW264.7 cells after LPS treatment.

### 3.5. In Silico Studies

Molecular docking is a computational modeling technique used to analyze or predict the stability and type of molecular interactions between ligands and receptors. The advantages of molecular docking are that it is relatively fast and cost-effective compared to experimental studies, and is becoming a trend in drug discovery or design development [66,67]. However, its limitations include the reliability of the scoring function and the requirement for experimental validation [68].

Lukito et al. [43] investigated the compounds contained in *A. paniculata* in a molecular docking test to screen drugs based on the highest bond affinity energy (∆G). The positive control used was Dexamethasone, and seven compounds with the lowest ∆G were obtained. This condition shows that the seven compounds are stable against CD14 protein binding. The seven compounds are andrographiside, andrographolactone, beta sitosterol, neoandrographolide, skullcapflavone I, 14-Deoxyandrographoside, and apigenin. An ensemble docking test was conducted for all seven to identify compounds that consistently have binding energy after five repetitions. Meanwhile, the assembly docking data are the average value of ΔG from 10 different structural conformations of the receptor-ligand complex obtained every 20 ns over a total span of 200 ns divided into 10 clusters. The ensemble docking result showed four stable compounds, namely, andrographiside, 14-Deoxyandrographoside, neoandrographolide, and apigenin, with ΔG less than −6.25 kcal/mol. In another study, neoandrographolide and andrographiside were reported to have strong binding to inflammatory receptors, including COX-2 (1CX2), and was used as treatment agents for COPD [69,70,71,72].

Astuti [44] worked in silico on the potential of *A. lavenia* and *Ageratum conyzoides* (*babadotan*) compounds for IL-6 protein receptors. The predicted results of IL-6 interaction with bioactive compounds in *A. lavenia* and *A. conyzoides* plants can be observed in Table 4 and Table 5. The compound *ent*-11α-hydroxy-15-oxo-kaur-16-en-19-oic acid contained in *A. lavenia* produced the highest probability score in interacting with IL-6. Meanwhile, the bioactive 22,23-Dihydrospinasterol and spinasterol found in *A. conyzoides* had the strongest interaction with IL-6. In another study, Kotta et al. [73] reported that β-sitosterol and stigmasterol have strong binding to the COX-2 (4 COX) receptor with binding energies of −8.9 kcal/mol and −7.4 kcal/mol, respectively. These two compounds were also reported by Astuti [44], as in Table 2. Furthermore, Vikasari et al. [56] identified that kaempferol-7-O-rhamnopyranoside (∆G = −9.57 kcal/mol) and kaempferol-3-O-glucoside (∆G = −9.54 kcal/mol) bind strongly to the P38-MAPK (mitogen-activated protein kinase) receptor.

Moreover, Tuwalaid et al. [47] analyzed compounds contained in *A. lavenia* and *Muntingia calabura* leaves against COX-2, TNF-α, IL-1β, and IL-6 receptors in silico. The specific receptors used in the four proteins are 5IKR, 2AZ5, 5R8E, and 1ALU (Figure 6a–d).

Based on the in silico analysis conducted by Tuwalaid et al. [47], there are three compounds that can inhibit all four inflammatory mediator proteins and pro-inflammatory cytokines from the extract of *A. lavenia*: 1a,9b-Dihydro-1H-cyclopropa[a]anthracene; biphenyl, 3,4-Diethyl; and 3,6-Dimethylphenanthrene. Meanwhile, there are four compounds that have the lowest binding energy to the four inflammatory proteins, namely, 1a,9b-Dihydro-1H-cyclopropa[A]anthracene; 3,6-Dimethylphenanthrene; and *ent*-11α-hydroxy-15-oxo-kaur-16-en-19-oic acid (Table 6). Furthermore, Tuwalaid et al. [47] also obtained data indicating that there are three compounds that can bind strongly to inflammatory receptors: 7-Hydroxyflavone to bind COX-2; β-Amyrenone to bind TNF-α and IL-1β; and Lupenone to bind IL-6 with the binding energy −9.1 kcal/mol; −9.2 kcal/mol; −9.1 kcal/mol; and −7.9 kcal/mol, respectively. The compounds with the lowest binding energy to four inflammatory mediator proteins and pro-inflammatory cytokines were 7-hydroxyflavone, β-amyrenone, and lupenone (Table 7). Interestingly, 8-hydroxy-6-methoxyflavone and diclofenac were shown by Chadalawada et al. [74] to bind strongly to the COX-2 receptor, corresponding with the report by Tuwalaid et al. [47], as shown in Table 7.

### 3.6. In Vivo Studies

In vivo studies in animal models, such as mice and rats, are frequently conducted to explore COPD and anti-inflammatory treatment due to the physiological relationship with humans. Controlled exposure to intranasal LPS and cigarette smoke is commonly used to mimic the pathogenesis of COPD. Meanwhile, anti-inflammatory investigations use localized inflammation such as paw edema caused by carrageenan to evaluate treatment efficacy. Indicators including TNF-α, IL-6, IL-17, and reduced tissue damage or edema are used to measure the effects of the extracts administered at calculated doses. The results obtained are verified with control groups, and ethical standards helped to ensure human care [75,76,77].

A pathological model simulating COVID-19-associated inflammation was generated using a method to increase IL-6 levels in lung tissue. The anti-inflammatory properties of the extract were assessed by quantifying IL-6 levels in lung tissue [78]. Kongsomros et al. [49] found that an ethanol extract of *A. paniculata* might be used to treat COVID-19. In hamsters, the extract decreased IL-6 levels in lung tissue (7278 ± 868.4 pg/g tissue) compared to the control (12,495 ± 1118 pg/g tissue) at loading doses of 1000 mg/kg and 500 mg/kg for seven days. Another study by Kevin et al. [79] reported that *A. paniculata* leaf extract reduced ICAM-1 (Intercellular adhesion molecule-1) and E-selection expression in the lung tissue of LPS-induced sepsis rats. At a concentration of 500 mg/kgBW, the extracts significantly lowered ICAM-1 (*p* < 0.001) and E-selection (*p* < 0.001). In COPD, ICAM-1 facilitates leukocyte adhesion and transmigration into lung tissue, worsening chronic inflammation, while E-selection mediates leukocyte rolling and recruitment during inflammatory responses. By downregulating these adhesion molecules, *A. paniculata* can potentially reduce leukocyte-driven inflammation and tissue damage in COPD.

Nurhasanah et al. [50] explored an *M. calabura* leaf extract capable of inhibiting the function of IL-17 in vivo as a candidate for COPD treatment using the cigarette smoke exposure method. The test used mice exposed to eight cigarettes for 60 min, and five times a week; then, LPS was given intranasally at weeks 1, 3, 5, and 7. Key outcomes, such as cytokine levels, mucus production, and lung tissue changes, were measured to assess the efficacy of the therapeutic agent. The test animals had not been exposed to smoke before receiving LPS. The 50% ethanol extract of *M. calabura* leaves (3.5 mg/20 g weight of mice) showed possible inhibition of cytokine IL-17, reduced mucus production, and lowered TNF-α expression in the lungs of COPD mouse models.

In vivo studies on *M. calabura*’s anti-inflammatory activities were also carried out by Widyaningrum et al. [51] through the administration of an inflammatory agent to generate localized edema. The reduction in edema volume was measured and compared to a positive control to benchmark the anti-inflammatory effects. At 240 mg/kg BW, the *M. calabura* ethanol extract showed a 62.51% decrease in edema volume (IEV), which was close to the positive control with *p*-value < 0.05 (acetosal = 52.12%), signifying 240 mg/kg BW as an effective dose. In the balm stick formulation, 10% of *M. calabura* extract reduced the edema volume in rat paws by 95.83% [52]. *M. calabura* leaf extract at 3.5 mg/20 g reduced IL-17a and TNF-α expression, as well as mucus production, in an in vivo COPD mice model induced by cigarette smoke and LPS. Furthermore, the methanol extract at 50 mg/kg significantly (*p* < 0.05) reduced TNF-α, IL-1β, and IL-6 in the tissue of Sprague Dawley rats [53]. The methanol extract from the fruits at doses of 100, 200, and 300 mg/kg significantly (*p* < 0.05) reduced carrageenan-induced edema in the hind paws of Wistar Albino rats over 3 h and showed no acute toxicity or lethality at doses of 1000 mg/kg, suggesting safety [54]. These results showed the potential of *M. calabura* as an anti-inflammatory agent, supporting the application for inflammatory conditions including COPD.

An aqueous leaf extract of *A. conyzoides* showed anti-inflammatory activity by reducing the expression of pro-inflammatory cytokines IL-1β, IL-18, TNF-α, and caspase-1 through the suppression of NOD-like receptor family pyrin domain containing 3 (NLRP3) inflammasome activation in a peritonitis mouse model [55]. In the acute anti-inflammatory assay conducted by Vikasari et al. [56], the *A. conyzoides* extract at doses of 45 and 90 mg/kg BW showed comparable anti-inflammatory effects (*p* < 0.05) to the positive control (diclofenac sodium, *p* > 0.05). The crude extract of *A. conyzoides*, as well as the ethanol, hexane, ethyl acetate, and dichloromethane fractions, significantly decreased the levels of pro-inflammatory cytokines, including IL-17A, IL-6, TNF, and IFN-γ (*p* < 0.05), in male Swiss mice induced with carrageenan [57]. The anti-inflammatory study on edema inhibition in rats reported by Galati et al. found that a 500 mg/kg dose of methanol extract of A. conyzoides achieved an inhibition percentage of 58.71 ± 0.0430% (*p* < 0.05) at 60 min [58].

Samsuar et al. [59] examined the leaves of *sembung rambat* (*Mikania micrantha* Kunth.). The 70% ethanol extract of *sembung rambat* leaves was tested for the ability to reduce inflammation-induced swelling of legs in male white rats. The dose variants given were 112.5 mg/kg, 225 mg/kg BW, and 450 mg/kg BW, while the positive control used Acetosal, and the negative control was administered with 1% CMC solution. The 450 mg/kgBB dose had better effects than the other two administered doses. The results showed that the ethanol fraction of the leaves of *sembung rambat* could inhibit swelling of the legs of male white rats of the Wistar strain. The greater the concentration of the fraction, the greater the inhibition of swelling in the legs. The area under the curve (AUC) is often used to quantify the effectiveness of treatment representing the cumulative measure of a response over time. A smaller AUC value signifies greater efficacy due to reflecting a reduced magnitude and duration of swelling [80], where the dose of 450 mg/kbBB has an average value of −8.22 mm.min. The study conducted by Deori et al. [60] reported that the ethanolic extract of *M. micrantha* leaves at doses of 200 and 400 mg/kg BW reduced acute inflammation in carrageenan-induced rat paw edema.

## 4. Conclusions

In conclusion, this systematic review showed that COPD treatment could be investigated by inflammation regulation through in vitro, in silico, and in vivo studies. Additionally, *sambiloto* (*A. paniculata*), *legetan warak* (*A. lavenia*), *kersen* (*M. calabura*), *babadotan* (*A. conyzoides*), and *sembung rambat* (*M. micrantha* Kunth.) were five Indonesian medicinal herbs identified with high potential as COPD treatment agents. These five plants have been proven in vitro to decrease pro-inflammatory cytokines such as TNF-α, IL-1β, and IL-6 in LPS-stimulated RAW 264.7 cells and inhibit the activity of inflammatory mediators (COX-2). In vivo research demonstrated that the five plants reduced inflammation in lung tissue in animal models. In in silico studies, compounds playing an important role in inhibiting inflammatory pathways included 14-Deoxyandrographoside, *ent*-11α-hydroxy-15-oxo-kaur-16-en-19-oic acid, 22,23-Dihydrospinasterol, and 7-Hydroxyflavone in *A. paniculata*, *A. lavenia*, *A. conyzoides*, and *M. calabura*, respectively. These were suspected to have a strong binding with inflammatory receptors, such as COX-2 (5IKR; 1CX2), TNF-α (2AZ5), IL-1β (5R8E), and IL-6 (1ALU).

## 5. Future and Prospects

Herbal treatment for COPD in Indonesia has great potential for development. Our research team is currently working on combinations of the five medicinal plants that we have reviewed in this systematic review. We have conducted multiple rounds of in vitro, in silico, and in vivo studies to provide scientific evidence for the treatment of COPD.

In the future, we want to work on a combination of pilot scale and collaborate with the pharmaceutical industry to develop herbal products at the *OHT* level (*Obat Herbal Tradisional*), or traditional herbal medicine recognized by Indonesia’s Food and Drug Supervisory Agency. We are also planning clinical data on this herbal combination so that our herbal products can be upgraded to the *phytopharmaca* level. This COPD research has received support from the Indonesian government through funding from the National Research and Innovation Agency. For that reason, we continue to achieve our research development goals.

## Figures and Tables

**Figure 1 cimb-47-00262-f001:**
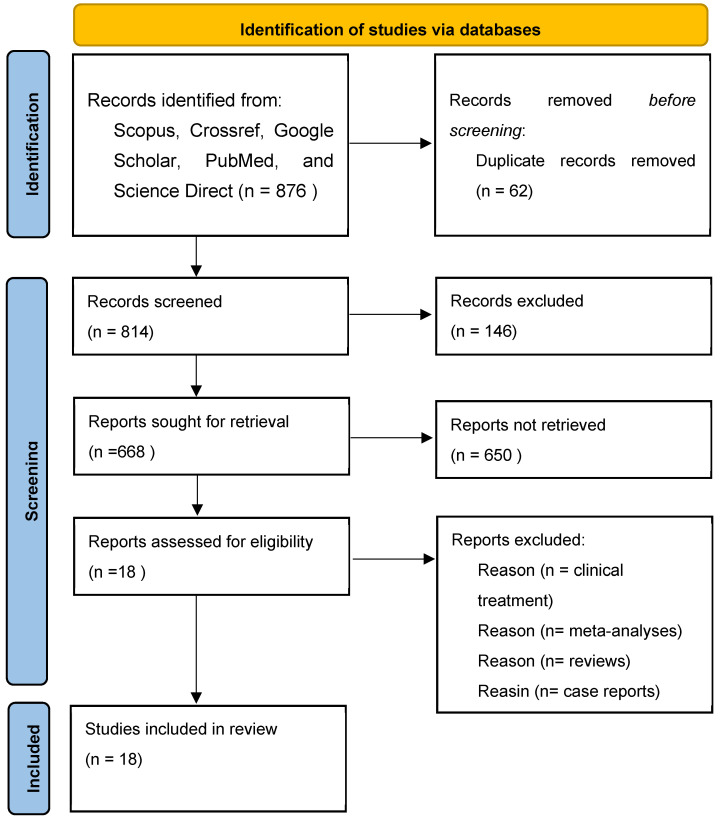
Flowchart for the selection of 18 studies included in this review.

**Figure 2 cimb-47-00262-f002:**
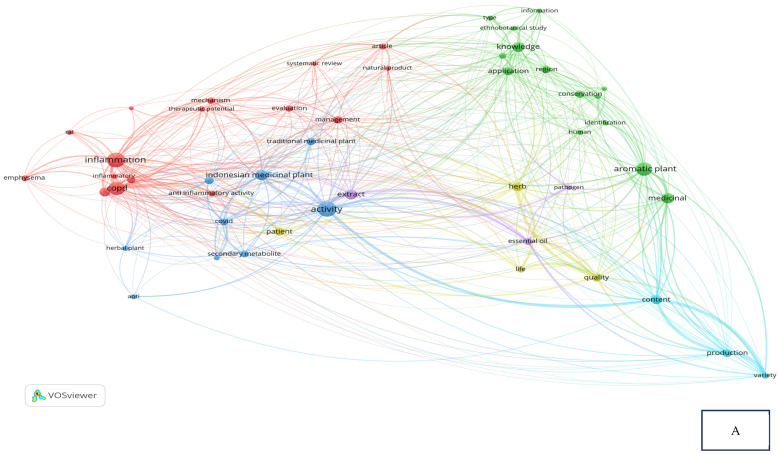
Network visualization (**A**), overlay visualization (**B**), and density visualization (**C**) of COPD and inflammation (the picture was generated from the VOSviewer software program, https://www.vosviewer.com/ (accessed on 19 February 2025)).

**Figure 3 cimb-47-00262-f003:**
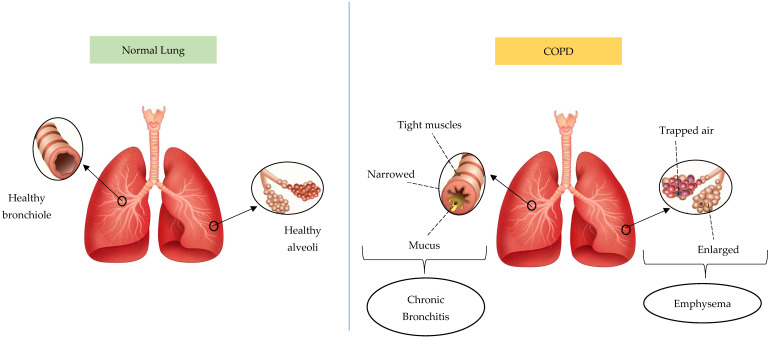
The difference between normal lungs and lungs affected by COPD. The pictures were generated from the free applications https://www.biorender.com/ (accessed on 20 February 2025) and https://id.pinterest.com/ (accessed on 20 February 2025). The authors also provide innovative pictures of normal lungs and COPD based on the systematic review that has been obtained.

**Figure 4 cimb-47-00262-f004:**
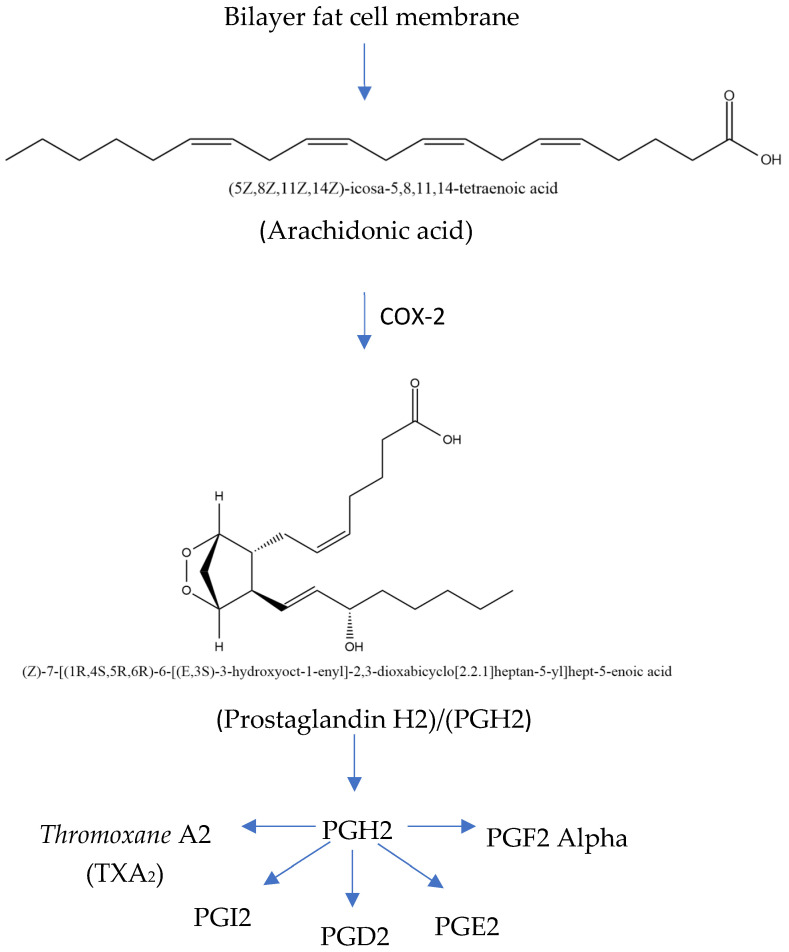
Mechanism of inflammatory pathway in COX-2 (modified from Kharwar et al. [41]; license number: 5938020036633).

**Figure 5 cimb-47-00262-f005:**
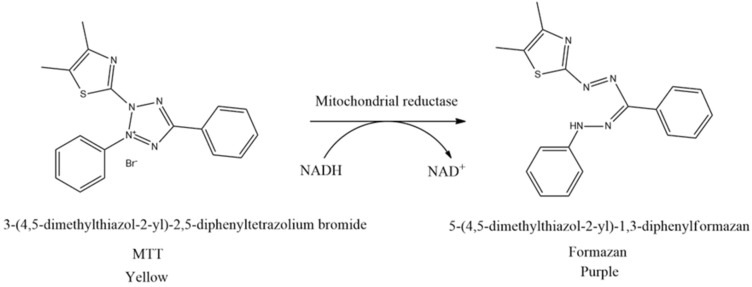
Reduction of MTT to formazan crystals.

**Figure 6 cimb-47-00262-f006:**
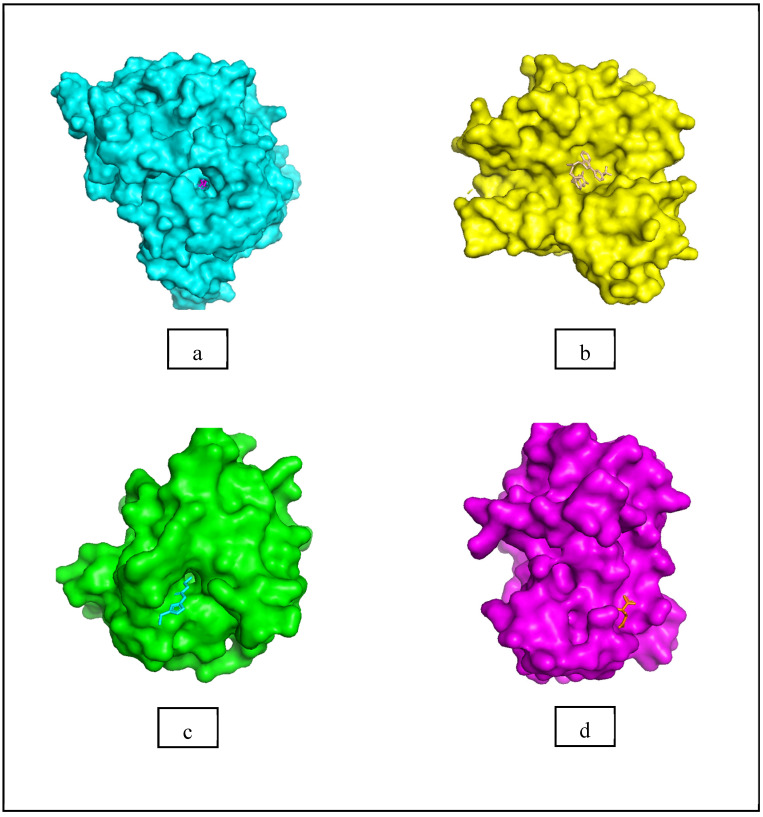
(**a**) 5IKR (Chain A); (**b**) 2AZ5 (Chain A and B); (**c**) 5R8E (Chain A); (**d**) 1ALU (Chain A). (the structures of proteins are from the Protein Data Bank https://www.rcsb.org/ (accessed on 20 February 2025) and the authors provide the structure visualizations from the PyMOL software program, https://www.pymol.org/ (accessed on 19 February 2025)).

**Table 1 cimb-47-00262-t001:** Inclusion and exclusion criteria.

Inclusion Criteria	Exclusion Criteria
Indonesian medicinal plants Inflammation Cytokine proinflammation in vitro, in vivo, in silico studies Articles published 2001–2024 Published in English	Meta-analysis Reviews (scoping, narrative, and systematic) Case reports, case series, and cross-sectional studies. Clinical data

**Table 2 cimb-47-00262-t002:** Bibliometric analysis: cluster and items based on VOSviewer result.

Number of Cluster	Number of Items	Items
1.	1.	Anti-inflammatory activity
	2.	Article
	3.	Chronic Obstructive Pulmonary Disease
	4.	Evaluation
	5.	Inflammation
	6.	Inflammatory
	7.	Mechanism
	8.	Selected medicinal plant
	9.	Therapeutic potential
2.	1.	Application
	2.	Aromatic plant
	3.	Conservation
	4.	Cultivation
	5.	Ethnobotanical study
	6.	Identification
	7.	Information
	8.	Medicinal
	9.	Region
3.	1.	Activity
	2.	Anti-inflammatory
	3.	COVID-19
	4.	Indonesian medicinal plant
	5.	Traditional medicinal plant
4.	1.	Life
	2.	Patient
	3.	Quality
5.	1.	Essential oil
6.	1.	Production
	2.	Variety

**Table 3 cimb-47-00262-t003:** General characteristics of selected studies.

No.	Aims of the Study	Methodology	Overall Findings	Reference
1.	To investigate the anti-proinflammatory cytokines of a 98% ethanol extract of *A. paniculata* using in vitro and in silico methods.	The process of molecular docking was used as an initial screening to assess the potential for suppressing macrophage cell activation. To examine the anti-cytokine pro-inflammatory activity in LPS-induced RAW 264.7 macrophage cells, TNF-α, IL-1ß, and IL-6 levels were assessed using an ELISA technique. Metabolite compounds were profiled using LC-MS/MS.	Approximately 100% of TNF-α and 85.59%of IL-1β were suppressed by ethanol extract containing specific metabolites 14-Deoxyandrographoside and 14-Deoxy-17-hydroxyandrographolide. Compound 14-Deoxyandrographoside showed a relatively high binding stability of −7.5 kcal/mol to the CD14 receptor.	Lukito et al. [43]
2.	To identify bioactive compounds that can inhibit IL-6 in *Adenostemma Lavenia* (*A. lavenia*) and *Ageratum conyzoides* (*A. conyzoides*) herbs; and to verify the findings of in silico evaluation with anti-inflammatory effects of *A. lavenia* herb extracts in vitro and *A. conyzoides* herb extracts to be developed as raw material for COPD herbal medicine.	Machine learning approach with a random fast method to predict drug-target interaction (DTI) based on chemical structure and genome sequence information. Plant extracts were examined for anti-inflammatory effects in RAW 264.7 macrophage cell cultures. The IL-6 levels were measured using an ELISA kit.	Ent-11α-hydroxy-15-oxo-kaur-16-en-19-oic acid in *A. lavenia*, as well as steroid and triterpenoid group compounds in *A. conyzoides*, suppress IL-6. The aqueous extract of *A. lavenia* can reduce IL-6 levels and does not affect the viability of inflammatory cells.	Astuti [44]
3.	To investigate phylogenic and biochemical of *A. lavenia* from Japan and Taiwan.	Analyzing chloroplast DNA sequences on *A. lavenia* from Japan and Taiwan. Measurement of protein levels by Keap1 (Kelch-like ECH-associated protein 1) method.	The study of kaurenoic acids showed that Japanese *A. lavenia* and *A. madurense* had high amounts of 11αOH-KA and moderate levels of 11α,15OH-KA, while Taiwanese *A. lavenia* mostly contained 9,11αOH-KA. These compounds can act as anti-inflammation (gene expression). Keap1 (protein levels were reduced by 11αOH-KA and 9,11αOH-KA. These compounds differ from 11α,15OH-KA by having a ketone (αβ-unsaturated carbonyl group, a thiol modulator) at the 15th position.	Maeda et al. [45]
4.	To modify the structure of Adenostemmoic acid B (AB) and to investigate the anti-inflammation activity of modified AB.	Through modifying the 19th position (carboxy: implicated in the prevention of cytotoxicity) of AB and to investigate the anti-inflammation of modified AB.	Long-chain alkylation of the carboxy group inhibited nitric oxide (NO) production and inducible NO synthase (iNOS) activity. Adenostemmoic acid B and the anhydride bond-containing derivatives suppress the iNOS enzyme activity.	Kobayashi et al. [46]
5.	To explore the potential utilization of plant extracts of *A. lavenia* and *Muntingia calabura* as anti-inflammatory agents in suppressing COX-2 activity using in vitro and in silico experiments.	Molecular docking and molecular dynamics modeling were applied to determine the stability of the complex between COX-2 and ligands. COX-2 inhibition was measured using the COX-2 Inhibitor Screening Assay KIT.	The compound from *A. lavenia*, 1a,9b-dihydro-1H-cyclopropa[a]anthracene, has the lowest binding energy (−8.7 kcal/mol). Meanwhile, *M. calabura* has a 7-hydroxyflavone ligand with a Gibbs free energy of −9.1 kcal/mol. The molecular dynamics analysis indicates that COX-2 retains its stability when forming contacts with chosen chemicals from all the studied extracts. The suppression of COX-2 assay revealed that a 96% EtOH extract of *A. lavenia* at concentrations of 25 and 100 ppm had 98% inhibitory activity. The 70% and 96% EtOH extracts of *M. calabura* at 1000 ppm concentration could inhibit COX-2 activity up to 100%.	Tuwalaid et al. [47]
6.	To use an in vitro method to examine the anti-inflammatory effects of *M. calabura*’s ethanolic extract	Determining the productions of pro-inflammatory mediators and protein expressions iNOS and COX-2 in LPS-stimulated RAW264.7 cells.	iNOS, COX-2, NO, and pro-inflammatory cytokines (TNF-α, IL-1β, and IL-6) in RAW264.7 macrophages can all be suppressed by *M. calabura* ethanolic extract.	Lin et al. [48]
7.	To examine the antiviral and anti-inflammatory potential of herbal extracts from *Andrographis paniculata* and *Boesenbergia rotunda*.	Applying a Golden Syrian hamster model infected with Delta, a representative variation connected to severe COVID-19.	*A. paniculata* extract administration significantly lowered IL-6 protein levels in lung tissue (7278 ± 868.4 pg/g tissue) compared to the control (12,495 ± 1118 pg/g tissue), implying decreased local inflammation. The extracts of *A. paniculata* and *B. rotunda* greatly lowered IL-6 and IP-10 mRNA expression in mononuclear cell populations constituting the peripheral blood of infected hamsters compared to the control group.	Kongsomros et al. [49]
8.	To investigate the connection between COPD pathophysiology and the potential of *kersen* (*M. calabura*) leaves, as well as analyze the mechanism of action of this plant as a therapeutic agent for treating COPD symptoms.	Molecular docking and molecular dynamics simulation were used to examine in silico study. The animals were male mice (Mus musculus). Additionally, an examination of IL-17A cytokines was performed through ELISA.	Molecular docking indicates that quercitrin has the lowest binding affinity value (−7.3 kcal/mol). In the lungs of COPD mouse models, *kersen*’s leaf extract at 3.5 mg/20 g of mouse weight demonstrated a potential suppression of the cytokine IL-17A, which decreased the generation of mucus.	Nurhasanah et al. [50]
9.	To investigate the anti-inflammatory properties of *M. calabura* L. leaves extract.	Using ethanolic, ethyl acetate, and chloroform solvents and observing a reduction in edema mice after 4 h of 1% carrageenan induction.	The ethanolic extract of *M. calabura* leaves exhibited anti-inflammatory properties. Since the results were nearly identical to the positive control (acetosal), the dose of 240 mg/kg BW was the most effective.	Widyaningrum et al. [51]
10.	To investigate the balm stick of *M. calabura* leaves extract’s topical anti-inflammatory properties.	Carrageenan-induced male white rats to investigate the anti-inflammatory effects of *M. calabura* leaves extract at different concentrations of 2.5%, 5%, and 10%.	Balm stick with a concentration of 10% extract demonstrated the highest anti-inflammatory action of more than 50% (95.83%).	Nugrahaeni et al. [52]
11.	To investigate the effect of the nitric oxide/cyclic-guanosine monophosphate (NO/cGMP) pathway in modulating the antinociceptive action of the methanol extract of *M. calabura* leaves (MEMC).	Mice (n = 6) were pretreated with 20 mg/kg L-arginine for 5 min before receiving 10% DMSO or MEMC (500 mg/kg). Mice were given an intraperitoneal injection of 0.6% acetic acid 60 min after receiving test solutions.	The peripheral and central processes are activated, and the NO/cGMP pathway and opioid receptors are partially modulated, as part of the MEMC antinociceptive implementation.	Sani et al. [53]
12.	To investigate the anti-inflammatory activity from the fruits of *M. calabura*.	In three hours, the methanolic fruit extracts decreased the edema caused by carrageenan in the hind paws of adult male Wistar Albino rats. The activity was contrasted with the activity of a common drug known as indomethacin.	The greatest percentage of inhibition (6243%) occurs at an extract concentration of 300 mg/kg, whereas the standard is 80.48%.	Preethi et al. [54]
13.	To identify how EAC (aqueous extracts of *A. conyzoides* leaf) works as an anti-inflammatory.	Monosodium urate (MSU)-induced peritonitis model was used to investigate the anti-inflammatory effects of EAC in vivo. The primary components of EAC were determined using ultra-performance liquid chromatography (UPLC) in conjunction with quadrupole-time-of-flight mass spectrometry (UPLC-Q-TOF-MS/MS).	The most effective components identified were kaempferol 3,7-diglucoside, 1,3,5-tricaffeoylquinic acid, and kaempferol 3,7,4′-triglucoside. EAC reduced the in vivo levels of inflammatory cytokines by inhibiting the activation of the NLRP3 inflammasome in a peritonitis mouse model.	Xu et al. [55]
14.	To determine the anti-inflammatory properties of eggshell membrane hydrolysates and *A. conyzoides* extract in vivo.	Using diclofenac-Na as a reference, the long-term anti-inflammatory effects of *A. conyzoides* extract and eggshell membrane hydrolysates were assessed in rats given cotton pellets.	Eggshell membrane hydrolysates and extract from *A. conyzoides* function in combination to reduce the severity of chronic inflammation.	Vikasari et al. [56]
15.	The crude extract (CE), its derived fractions (ethanol (EtOH-F), hexane (HEX-F), ethyl acetate (EtOAc-F), and dichloromethane (DCM-F), as well as isolated compounds like 5′-methoxy nobiletin (MeONOB), 1,2-benzopyrone, and eupalestin, which are obtained from the aerial parts of *A. conyzoides* L., were evaluated for their anti-inflammatory properties.	Using an animal model of carrageenan-induced inflammation. The five main inflammatory markers examined were TNF-α, IL-10, IL-17A, IL-6, and leukocyte inflow.	Isolated compounds are substantially reduced TNF-α, IL-17A, and IL-6.	Vigil de Mello et al. [57]
16.	To determine the effect of methanol extract of *A. conyzoides* and flavonoid fraction from the plant’s aerial component on carrageenan-induced edema in rats.	Edema was generated in each rat’s right hind paw by a subplantar injection of 0.05 mL of a 1% carrageenan suspension. The methanol extract and flavonoid fraction were provided 1 h before carrageenan injection, at a dose equivalent to 500 mg/kg of dried medication. A water plethysmometer was used to measure the volume of each paw.	The methanol extract of *A. conyzoides* inhibits 58.71% ± 0.0430 carrageenan-induced sub-plantar edema in rats for 60 min, whereas flavonoid fraction is 59.97% ± 0.0187.	Galati et al. [58]
17.	To investigate the implications of ethanol fraction on *sembung rambat* (*Mikania micrantha* Kunth) as an anti-inflammatory on male white rats of the wistar strain.	Following the extraction, additional fractionation is performed. The resulting fraction was subsequently administered at three doses: 112.5 mg/kgBW, 225 mg/kgBW, and 450 mg/kgBW.	Male white rats of the wistar strain showed anti-inflammatory effects from the ethanol portion of caustic granules. The optimal dosage is 450 mg/kgBW, with an average area under the curve (AUC) of 11.22 mm.s.	Samsuar et al. [59]
18.	To determine the anti-inflammatory properties of the ethanolic extracts of *Mikania micrantha* leaves (EEMM) on experimental animals with adjuvant-induced chronic arthritis, granuloma-pouch technique, and carrageenan-induced rat paw edema.	Using 95% alcohol, the percolation method was used to prepare the extract of *Mikania micrantha* leaves. Freund’s full adjuvant-induced arthritis method was used to study chronic inflammation, Granuloma pouch method for sub-acute inflammation, and carrageenan-induced rat paw oedema method for acute inflammation.	EEMM at doses of 200–400 mg/kg significantly reduced paw oedema in carrageenan-induced acute and sub-acute inflammation compared to control (*p* < 0.05). EEMM works in the chronic arthritis model in a dose-dependent way.	Deori et al. [60]

Abbreviation: TNF-α: Tumor Necrosis Factor-alpha. IL-1β: Interleukin-1 beta. IL-6: Interleukin-6. IL-17: Interleukin-17. IL-10: Interleukin-10. COX-2: cyclooxygenase-2. LPS: Lipopolysaccharide. ELISA: Enzyme-linked immunosorbent assay. DTI: drug-target interaction. NO: nitric oxide. iNOS: inducible NO synthase. NO/cGMP: nitric oxide/cyclic-guanosine monophosphate. MSU: Monosodium urate. UPLC-Q-TOF-MS/MS: Ultra-Performance Liquid Chromatography-Quadrupole-Time-of-Flight Mass Spectrometry.

**Table 4 cimb-47-00262-t004:** Prediction results of IL-6 interaction with *A. lavenia* bioactive compounds (Astuti [44]).

No.	Compounds	Probability Score	Chemical Structure
1.	*Ent*-11α-hydroxy-15-oxo-kaur-16-en-19-oic acid	0.688690476	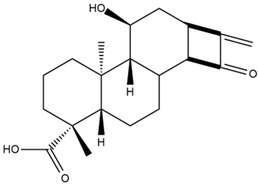
2.	4-*O*-[3-acetyl-1-(trimethylsilyl)-1h-indolyl]-D-glucose	0.616309524	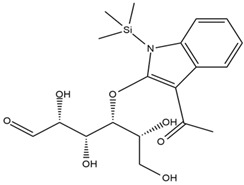
3.	6,7-Dihydro-3-nitro-5h-cyclopenta[b]pyridine-2(1h)-one	0.579642857	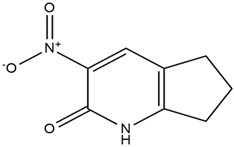
4.	5H-cyclopenta[b]pyridine	0.530434343	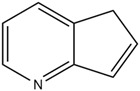

**Table 5 cimb-47-00262-t005:** Prediction results of IL-6 interaction with bioactive compounds in *A. conyzoides* (Astuti [44]).

No.	Compounds	Probability Score	Chemical Structure
1.	22,23-Dihydrospinasterol	0.7262	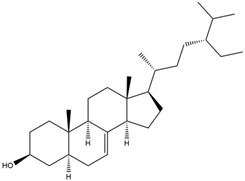
2.	Spinasterol	0.7248	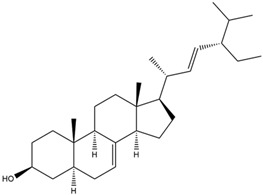
3.	β-sitosterol	0.7112	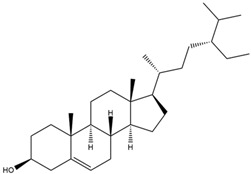
4.	Brassicasterol	0.7098	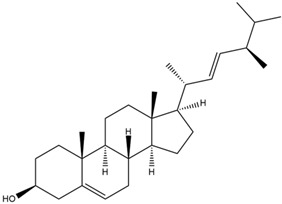
5.	Stigmasterol	0.7098	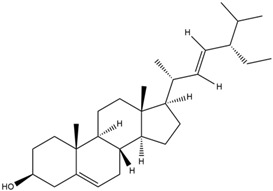

**Table 6 cimb-47-00262-t006:** Molecular docking results of *A. lavenia* compounds against COX-2; TNF-α; IL-1β; and IL-6 (Tuwalaid et al. [47]).

No.	Compounds	Receptor 1		Receptor 2		Receptor 3		Receptor 4		Chemical Structure
		COX-2	Binding Energy (kcal/mol)	TNF-α	Binding Energy (kcal/mol)	IL-1β	Binding Energy (kcal/mol)	IL-6	Binding Energy (kcal/mol)	
1	1a,9b-Dihydro-1H-cyclopropa[a]anthracene	✓	−8.7	✓	−7.7	✓	−6.9	✓	−6.2	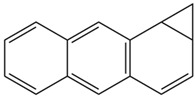
2	Biphenyl, 3,4-Diethyl	✓	−8.7	✓	−7.4	✓	−6.2	✓	−6.2	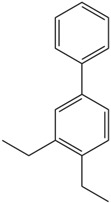
3	3,6-Dimethylphenanthrene	✓	−8.3	✓	−7.8	✓	−6.9	✓	−6.5	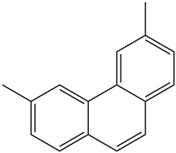
4	Diclofenac	✓	−8.1	ND	ND	ND	ND	ND	ND	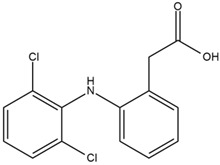
5	Linoleic acid	✓	−7.0	ND	ND	ND	ND	ND	ND	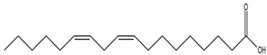
6	*ent*-11α-hydroxy-15-oxo-kaur-16-en-19-oic acid	ND	ND	✓	−9	✓	−7.1	✓	−7.1	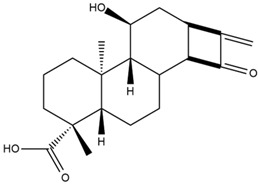
7	1-cyano-3-methylisoquinoline	ND	ND	✓	−7.5	✓	−6.2	✓	−6	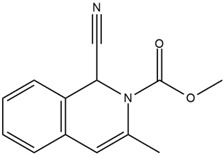

ND: not determined.

**Table 7 cimb-47-00262-t007:** Molecular docking results of *M. calabura* compounds against COX-2; TNF-α; IL-1β; and IL-6. (Tuwalaid et al. [47]).

No.	Compounds	Receptor 1		Receptor 2		Receptor 3		Receptor 4		Chemical Structure
		COX-2	Binding Energy (kcal/mol)	TNF-α	Binding Energy (kcal/mol)	IL-1β	Binding Energy (kcal/mol)	IL-6	Binding Energy (kcal/mol)	
1	7-Hydroxyflavone	✓	−9.1	ND	ND	ND	ND	ND	ND	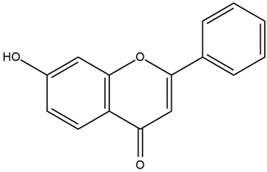
2	7-Hydroxyisoflavone	✓	−9.0	ND	ND	ND	ND	ND	ND	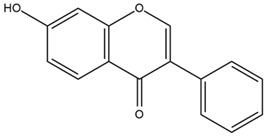
3	8-Hydroxy-6-methoxyflavone	✓	−9.0	ND	ND	ND	ND	ND	ND	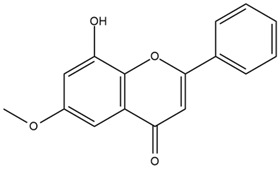
4	(2S)-7-Hydroxyflavanone	✓	−8.9	ND	ND	ND	ND	✓	−6.7	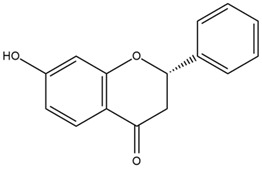
5	5,7-Dihydroxyflavone	✓	−8.9	ND	ND	ND	ND	ND	ND	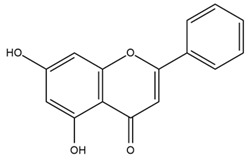
6	Diclofenac	✓	−8.1	ND	ND	ND	ND	ND	ND	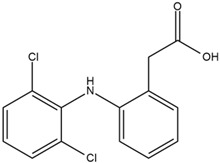
7	β-Amyrenone	ND	ND	✓	−9.2	✓	−9.1	✓	−7.8	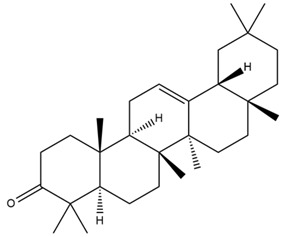
8	Lupenone	ND	ND	✓	−9.1	✓	−9.0	✓	−7.9	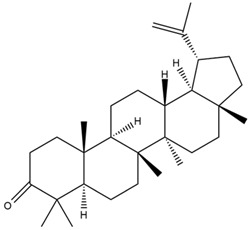
9	Stigmasterol	ND	ND	✓	−9.0			✓	−7.1	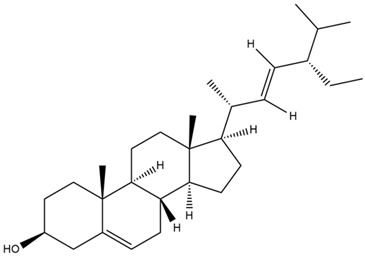
10	2alpha,3beta-Dihydroxy-olean-12-en-28-oic acid	ND	ND	✓	−8.5	✓	−8.2	✓	−7.5	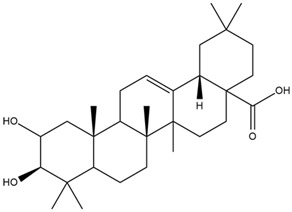
11	β-Sitosterol	ND	ND	✓	−8.4	ND	ND	ND	ND	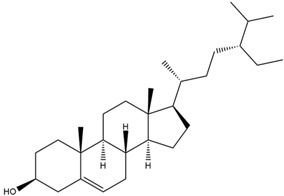
12	β-Sitostenone	ND	ND	ND	ND	✓	−7.5	ND	ND	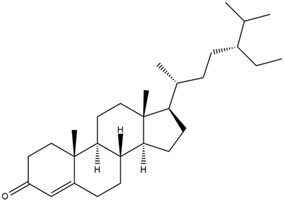
13	(2S)-5-Hydroxy-7-methoxyflavanone	ND	ND	ND	ND	✓	−7.4	ND	ND	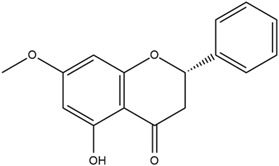

ND: not determined.

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
