# Peer review of "Inhibition of Inflammatory Regulators for Chronic Obstructive Pulmonary Disease (COPD) Treatment from Indonesian Medicinal Plants: A Systematic Review"

_cimb, 2025, doi:10.3390/cimb47040262_

Round 1

Reviewer 1 Report

Comments and Suggestions for Authors I have reviewed the submission entitled “Inhibition of Inflammatory Regulation for COPD Treatment from Indonesian Medicinal Plants: A Systematic Review” for Current Issues in Molecular Biology. After a thorough review, this version of the submission cannot move forward at this time, because: 1. The manuscript presents a systematic review of Indonesian medicinal plants for COPD treatment; however, it does not provide significant new insights. The findings primarily reiterate existing knowledge without establishing a unique contribution to the field. 2. The systematic review does not fully adhere to PRISMA guidelines. The inclusion and exclusion criteria are not clearly defined, and there is insufficient discussion on the reliability and validity of the sources reviewed. Furthermore, the rationale for selecting specific medicinal plants lacks adequate justification. 3. The discussion section lacks an in-depth appraisal of study limitations, potential biases, and existing research gaps. A more comprehensive and critical analysis would enhance the scientific impact of the review. 4. The data synthesis and visualization methods (e.g., network and density visualization) require clearer explanations. Several tables and figures lack adequate textual support, making it difficult for readers to interpret the findings effectively. 5. The manuscript contains grammatical errors and unclear sentence structures that hinder readability. A professional language review is recommended to enhance clarity and coherence.

Author Response

Authors thanks for the opportunity to revise the manuscript. We are very appreciative of all of the comments and suggestions that useful for improving the quality of the manuscript. We have tried our best to address your comments point by point on your valuable concern. All of the revised parts have been presented in yellow highlight. We expect by this revised version, the manuscript can be reconsidered to fit with a high standard of the journal of Current Issues in Molecular Biology (CIMB).

Comments 1: I have reviewed the submission entitled “Inhibition of Inflammatory Regulation for COPD Treatment from Indonesian Medicinal Plants: A Systematic Review” for Current Issues in Molecular Biology. After a thorough review, this version of the submission cannot move forward at this time, because: 1. The manuscript presents a systematic review of Indonesian medicinal plants for COPD treatment; however, it does not provide significant new insights. The findings primarily reiterate existing knowledge without establishing a unique contribution to the field. 2. The systematic review does not fully adhere to PRISMA guidelines. The inclusion and exclusion criteria are not clearly defined, and there is insufficient discussion on the reliability and validity of the sources reviewed. Furthermore, the rationale for selecting specific medicinal plants lacks adequate justification. 3. The discussion section lacks an in-depth appraisal of study limitations, potential biases, and existing research gaps. A more comprehensive and critical analysis would enhance the scientific impact of the review. 4. The data synthesis and visualization methods (e.g., network and density visualization) require clearer explanations. Several tables and figures lack adequate textual support, making it difficult for readers to interpret the findings effectively. 5. The manuscript contains grammatical errors and unclear sentence structures that hinder readability. A professional language review is recommended to enhance clarity and coherence

Response 1: Dear reviewer, thank you very much for the suggestions and comments.

1)      The uniqueness of our systematic review has been added and explored in the introduction section on page 2 Lines 67─85. Treatment of Chronic Obstructive Pulmonary Disease (COPD) can be done through an inflammatory pathway approach and 5 Indonesian medicinal plants that we reviewed have been studied in vitro, in silico, and in vivo metods and have been proven to reduce inflammatory activity on COX-2, TNF-α, IL-1β, and IL-6 receptors.

2)      We have provided and revised the systematic review through registered in PROSPERO with ID CRD420250645213, adding the search strategy, adding inclusion and exclusion criteria. We also added the source or software program that we used for generating all related papers. We can observe on page 3 Lines 97─179.

3)      In the results and discussions, we have explored the connection between COPD and inflammation, and we have provided the summary on 18 selected studies in Table 3. We have added the future and prospects for our responsibility in our following work on Lines 676─688.

4)      This systematic review has been added in bibliometric analysis to make critical review the gap analysis and new insight in developing COPD treatments using Indonesian medicinal plants. Lines 181─246. The figures and tables also we added the chemical structure of the compounds. We can observe in Lines 510─512; Lines 530─532; Lines 563-565; Lines 578-579.

5)      We have improved our English grammar via GoodLingua proofreading and we have attached the certificate. We really hope our revisions can be well-received.

Reviewer 2 Report

Comments and Suggestions for Authors

After a thorough review of the manuscript entitled “Inhibition of Inflammatory Regulation for COPD Treatment from Indonesian Medicinal Plants: A Systematic Review”, I highlighted some points that should be taken into consideration by the authors to improve the study.

1. The meaning of the acronym COPD should be included in the manuscript title.

2. In the abstract, authors should include the main results on the anti-inflammatory potential of Indonesian medicinal plants.

Furthermore, authors must make it clear in the abstract that this study is a systematic review and specify the number of articles included.

3. Line 24: It is necessary to replace the keywords with others that better cover the content described in this manuscript.

4. There are several previously published review studies that have reported the anti-inflammatory potential of Indonesian medicinal plants. In this context, the authors should include information in the introduction of this manuscript about the use of medicinal plants for the treatment of inflammatory diseases in Indonesia. It is important to mention the therapeutic properties of the five species targeted by this study (Andrographis paniculata, Adenostemma lavenia, Muntingia calabura, Ageratum conyzoides, and Mikania micranta).

5. Line 103: In the methodology, authors must specify the chronological period determined to carry out searches for articles in the databases.

6. Line 110: Authors must clearly specify the criteria used for inclusion of articles in this review study.

7. In the methodology text, the authors reported that 12 studies were included in this review (Line 115). However, in the caption to Figure 2, the authors reported the inclusion of 18 articles (Line 118). This is confusing.

Furthermore, in the representation of the “Included” stage in Figure 2, there is no value in “(n = )”.

8. Figure 3: It is necessary to specify whether the authors themselves created this illustration.

9. Table 1: I suggest that the authors reduce the information in Table 1. There is a lot of text that can be summarized to improve the visualization in the table.

10. Tables 2, 3, 4 and 5: It is necessary to include the chemical structure drawing of all compounds mentioned in this manuscript.

11. Authors should include a new section in this manuscript entitled “Gaps and Future Perspectives.” Based on the analysis of the articles included in this review, it is necessary to clarify the main flaws found in the works and suggest new perspectives related to the use of Indonesian medicinal plants for the treatment of COPD.

Author Response

Authors thanks for the opportunity to revise the manuscript. We are very appreciative of all of the comments and suggestions that useful for improving the quality of the manuscript. We have tried our best to address your comments point by point on your valuable concern. All of the revised parts have been presented in yellow highlight. We expect by this revised version, the manuscript can be reconsidered to fit with a high standard of the journal of Current Issues in Molecular Biology (CIMB).

Comments 1: The meaning of the acronym COPD should be included in the manuscript title.

Response 1: Thank you so much for the suggestion. We have added the acronym COPD in the title of manuscript. We can observe in Lines 1─4.

Comments 2: In the abstract, authors should include the main results on the anti-inflammatory potential of Indonesian medicinal plants. Furthermore, authors must make it clear in the abstract that this study is a systematic review and specify the number of articles included.

Response 2: Thank you so much for the suggestions. We have revised the abstract that this study is a systematic review and the number of articles. We also added the main result of the study. We can observe in Lines 22─23.

Comments 3: Line 24: It is necessary to replace the keywords with others that better cover the content described in this manuscript

Response 3: Thank you so much for the suggestions. We have added one new keyword: systematic review. We can observe in Line 35.

Comments 4: There are several previously published review studies that have reported the anti-inflammatory potential of Indonesian medicinal plants. In this context, the authors should include information in the introduction of this manuscript about the use of medicinal plants for the treatment of inflammatory diseases in Indonesia. It is important to mention the therapeutic properties of the five species targeted by this study (Andrographis paniculata, Adenostemma lavenia, Muntingia calabura, Ageratum conyzoides, and Mikania micranta).

Response 4: Thank you so much for the suggestions. We have added the explanation about using these five species Indonesian medicinal plants in COPD treatment. We can obersrve in Lines 66─85.

Comments 5: Line 103: In the methodology, authors must specify the chronological period determined to carry out searches for articles in the databases.

Response 5: Thank you so much for the suggestions. We have added the period time in reviewed paper and the databases or softaware program we used. We can obersrve in Lines 102─111.

Comments 6: Line 110: Authors must clearly specify the criteria used for inclusion of articles in this review study.

Response 6: Thank you so much for the suggestions. We have added the criteria used for inclusion and exclusion criteria in the review. We can obersrve in Lines 112─128.

Comments 7: In the methodology text, the authors reported that 12 studies were included in this review (Line 115). However, in the caption to Figure 2, the authors reported the inclusion of 18 articles (Line 118). This is confusing. Furthermore, in the representation of the “Included” stage in Figure 2, there is no value in “(n = )”

Response 7: Thank you so much for the correction. We have repaired the number studies we used. We can observe in Line 124, Line 174.

Comments 8: Figure 3: It is necessary to specify whether the authors themselves created this illustration.

Response 8: Thank you so much for your suggestion. We have provided a description of the innovative image we created. We can observe in Lines 323─324.

Comments 9: Table 1: I suggest that the authors reduce the information in Table 1. There is a lot of text that can be summarized to improve the visualization in the table.

Response 9: Thank you so much for the suggestions. However, we want to provide a comprehensive table in our 18 studies and some reviewers want this comprehensive analysis.

Comments 10: Tables 2, 3, 4 and 5: It is necessary to include the chemical structure drawing of all compounds mentioned in this manuscript.

Response 10: Thank you so much for the suggestions. We have added the chemical structure drawn on all compounds in Tables 4, 5, 6, and 7. We have used ChemDraw program. We can observe in Lines 510─512; Lines 530─532; Lines 563─565; Lines 578─579.

Comments 11: Authors should include a new section in this manuscript entitled “Gaps and Future Perspectives.” Based on the analysis of the articles included in this review, it is necessary to clarify the main flaws found in the works and suggest new perspectives related to the use of Indonesian medicinal plants for the treatment of COPD.

Response 11: Thank you so much for the suggestions. We have one new section about Future and Prospects. We can observe in Lines 676─688.

Reviewer 3 Report

Comments and Suggestions for Authors

we would like to acknowledge the authors for this very interesting article. However many points have to be change to enhance the quality of this paper:

  1. line 04: (1) the numbering has to follow the order, ( instead of 3 use the number 2 here). (2) add a comma between the number and the *.
  2. line 14: please correct the text size and police.
  3. line 15 to 23: try to put the text size and character in the size recommanded by the template.
  4. many statements in this document have to be referenced.
  5. correct line 29, 33, 47, 265, 266, 277, 300.
  6. line 54: correct the citation style.
  7. lines 57 to 61: this part have to move to results and discussion (bibliometric analysis).
  8. describe all the software used in this paper on the materials and methods section; add the reference of the softwares (company, country); add the last access date of the websites.
  9. put the figures and tables right after their mention in the text.
  10. correct the indicated titles.
  11. the chemical structures have to be drawn with chemdraw or marvinsketch (please improve their quality).
  12. enlarge the big tables (use the big tables from the MDPI template.
  13. Table 1: remove the author column, and format the references in the table to be in form of "Author et al. [number]"; after the table: down the table add the abreviations used in the table as well as what they means.
  14. Tables 4 & 5: for the empty cells please put ND (not determined).
  15. develop the conclusion and highlight the importance of you papers and its significance for future studies.

Author Response

Authors thanks for the opportunity to revise the manuscript. We are very appreciative of all of the comments and suggestions that useful for improving the quality of the manuscript. We have tried our best to address your comments point by point on your valuable concern. All of the revised parts have been presented in yellow highlight. We expect by this revised version, the manuscript can be reconsidered to fit with a high standard of the journal of Current Issues in Molecular Biology (CIMB).

Comments 1: line 04: (1) the numbering has to follow the order, ( instead of 3 use the number 2 here). (2) add a comma between the number and the *.

Response 1: Thank you so much for the correction. We have revised the numbering of affiliations. We can observe in Lines 5─6.

Comments 2: line 14: please correct the text size and police.

Response 2: Thank you so much for the correction. However, we have used the CIMB template, in the template of CIMB: the text size of affiliation is 8 and Palatino Linotype font.

Comments 3: line 15 to 23: try to put the text size and character in the size recommanded by the template.

Response 3: Thank you so much for the correction. However, we have used the CIMB template, in the template of CIMB: the text size of abstract is 9 and Palatino Linotype font.

Comments 4: many statements in this document have to be referenced.

Response 4: Thank you for the suggestions. However, we have used references in statements and sentences.

Comments 5: correct line 29, 33, 47, 265, 266, 277, 300.

Response 5: Thank you for the comments. However, could you give the explanation about the specific correct line.

Comments 6: line 54: correct the citation style.

Response 6: Thank you for the correction. We have used the citation style of CIMB template.

Comments 7: lines 57 to 61: this part have to move to results and discussion (bibliometric analysis)

Response 7: Thank you so much for the suggestions. We have added one new section in results and discussions about Bibliometric analysis. We can observe in Lines 182─245.

Comments 8: describe all the software used in this paper on the materials and methods section; add the reference of the softwares (company, country); add the last access date of the websites.

Response 8: Thank you for the suggestions. We have added the software program we used in search strategy. We can observe in Lines 102─111.

Comments 9: put the figures and tables right after their mention in the text.

Response 9: Thank you for the suggestions. We have revised the position of tables and figures after their mention in the text. We can observe in Lines 563─565; Lines 578─579.

Comments 10: correct the indicated titles

Response 10: Thank you for the suggestion. We have revised our title manuscript. We can oberve in Lines 1─4.

Comments 11: the chemical structures have to be drawn with chemdraw or marvinsketch (please improve their quality).

Response 11: Thank you for the suggestions. We have drawn the chemical structure with ChemDraw. We can observe in 327─362; Lines 426─427; Lines 510─512;  Lines 530─532; Lines  563─565; Lines 578─579.

Comments 12: enlarge the big tables (use the big tables from the MDPI template.

Response 12: Thank you for the suggestions. We have used the big tables in MDPI template.

Comments 13: Table 1: remove the author column, and format the references in the table to be in form of "Author et al. [number]"; after the table: down the table add the abreviations used in the table as well as what they means.

Response 13: Thank you for the suggestions. We have revised the Table 3 and added the abbreviations in bottom table. We can observe in Lines 380─395.

Comments 14: Tables 4 & 5: for the empty cells please put ND (not determined).

Response 14: Thank you for the suggestions. We have put ND for the empty cells. We can observe in Table 6 Lines 563─565; Table 7 Lines 578─579.

Comments 15: develop the conclusion and highlight the importance of you papers and its significance for future studies.

Response 15: Thank you for the suggestions. We have revised the conclusions and we have added one new section about Future and Prospects. We can observe in Lines 665─675; Lines 676─688.

Reviewer 4 Report

Comments and Suggestions for Authors

The manuscript discusses the use of traditional medicinal plants from Indonesia, their bioactivities, and importance in the development of medicine for Chronic Obstructive Pulmonary Disease (COPD).

Medicinal plants have been traditionally used to treat human ailments since ancient times and are gaining recognition for the development of natural products as drug molecules.

I have a few suggestions for the improvement of the manuscript:

The paper title needs to be modified. It should be rewritten as

Inhibition of Inflammatory Regulators for COPD Treatment from Indonesian Medicinal Plants: A Systematic Review

Abstract: The abstract needs to be reorganized by a native English speaker. In addition to providing a brief, and to the point abstract, the authors need to discuss the prospects and future directions of the study.

Research studies have highlighted the emerging importance of medicinal plants in the treatment of human disorders. What is the future outcome of the article? Discuss.

In addition to these 5 Indonesian medicinal plants, are there any other plant species effective for COPD? In the introduction section, key examples of other medicinal plants may be included.

The concept and execution of the literature review is systematic and clear, however, English language of the paper needs to be substantially improved for clarity and consistency.

Diagrammatic presentations are clear and well-drawn.

Table 1. Provides key information on selected case studies but has a lot of discussion. Will it possible to revise table 1 to discuss the content briefly? Please revise. In addition, the table will seem better in a landscape format. There should be a little gap between the two rows as to not mix the information.

Table 4. Molecular docking results of A. lavenia compounds. Please also include the reference/paper from where the table is taken. Similarly for table 5.

Comments on the Quality of English Language

Substantial English revisions are required for improved clarity and consistency of the content.

Author Response

Authors thanks for the opportunity to revise the manuscript. We are very appreciative of all of the comments and suggestions that useful for improving the quality of the manuscript. We have tried our best to address your comments point by point on your valuable concern. All of the revised parts have been presented in yellow highlight. We expect by this revised version, the manuscript can be reconsidered to fit with a high standard of the journal of Current Issues in Molecular Biology (CIMB).

Comments 1: The paper title needs to be modified. It should be rewritten as

Inhibition of Inflammatory Regulators for COPD Treatment from Indonesian Medicinal Plants: A Systematic Review

Response 1: Thank you so much for the suggestions. We have changed the title as you recommended. We can observe in Lines 1─4.

Comments 2: Abstract: The abstract needs to be reorganized by a native English speaker. In addition to providing a brief, and to the point abstract, the authors need to discuss the prospects and future directions of the study.

Response 2: Thank you so much for the suggestions. We have revised the abstract and  improved the English grammar via proofreading GoodLingua.

Comments 3: Research studies have highlighted the emerging importance of medicinal plants in the treatment of human disorders. What is the future outcome of the article? Discuss.

Response 3: Thank you so much for the suggestions. We have added one new section about Future and Prospects. We can observe in Lines 676─688.

Comments 4: In addition to these 5 Indonesian medicinal plants, are there any other plant species effective for COPD? In the introduction section, key examples of other medicinal plants may be included.

Response 4: Thank you for the suggestions. We have added more explanation about using these 5 Indonesian medicinal plants in the introduction. We can observe in Lines 67─85.

Comments 5: The concept and execution of the literature review is systematic and clear, however, English language of the paper needs to be substantially improved for clarity and consistency.

Response 5: Thank you so much for the suggestions. We have improved the grammar via proofreading GoodLingua and we attached the certificate.

Comments 6: Diagrammatic presentations are clear and well-drawn.

Response 6: Thank you so much for your appreciation.

Comments 7: Table 1. Provides key information on selected case studies but has a lot of discussion. Will it possible to revise table 1 to discuss the content briefly? Please revise. In addition, the table will seem better in a landscape format. There should be a little gap between the two rows as to not mix the information.

Response 7: Thank you so much for the suggestions. We have revised Table 2 in the landscape format. However, we want to provide a comprehensive table in our 18 studies. Some reviewers also want this comprehensive analysis.

Comments 8: Table 4. Molecular docking results of A. lavenia compounds. Please also include the reference/paper from where the table is taken. Similarly for table 5.

Response 8: Thank you for the suggestions. We have added the reference in Table 4─Table 7. We can observe in Lines 510─512; Lines 530─532; Lines 563-565; Lines 578-579.

Round 2

Reviewer 2 Report

Comments and Suggestions for Authors

The authors responded to my comments and necessary changes were made to the manuscript.

Reviewer 3 Report

Comments and Suggestions for Authors

I would like to extend my sincere appreciation for the thorough revisions you have made to your manuscript. Your responsiveness to the feedback provided has significantly improved the clarity and quality of your work.

Reviewer 4 Report

Comments and Suggestions for Authors

I appreciate the efforts made by the authors to improve the manuscript. It can be accepted in the present form.